# ADVANCING SUPERVISED LOCAL LEARNING BEYOND CLASSIFICATION WITH LONG-TERM FEATURE BANK

## ABSTRACT

Local learning offers an alternative to traditional end-to-end back-propagation in deep neural networks, significantly reducing GPU memory usage. While local learning has shown promise in image classification tasks, its application to other visual tasks remains limited. This limitation arises primarily from two factors: 1) architectures tailored for classification are often not transferable to other tasks, leading to a lack of reusability of task-specific knowledge; 2) the absence of cross-scale feature communication results in degraded performance in tasks such as object detection and super-resolution. To address these challenges, we propose the Memory-augmented Auxiliary Network (MAN), which introduces a simplified design principle and incorporates a feature bank to enhance cross-task adaptability and communication. This work represents the first successful application of local learning methods beyond classification, demonstrating that MAN not only conserves GPU memory but also achieves performance on par with end-to-end approaches across multiple datasets for various visual tasks.

## 1 INTRODUCTION

Back-propagation (BP) remains the cornerstone of deep learning optimization, but as models scale to larger sizes Bengio et al. (2006); Krizhevsky et al. (2017), End-to-End (E2E) methods expose several limitations Hinton et al. (2006); Guo et al. (2020). BP relies on the propagation of error signals across multiple layers, a process that contrasts with biological neural transmission systems Crick (1989) and introduces challenges, such as error accumulation in deep networks. This can degrade the learning effectiveness of shallow neurons Qu et al. (1997). Moreover, updating hidden layers in the deep network requires the completion of forward and backward passes, which hinders parallel computation and significantly increases memory consumption on GPUs Jaderberg et al. (2017); Belilovsky et al. (2020). As an alternative to E2E methods, supervised local learning enhances memory efficiency and parallelism by splitting the network into gradient-isolated blocks, each updated independently via its own auxiliary network Belilovsky et al. (2020); Nøkland & Eidnes (2019).

However, current applications of local learning have largely been confined to image classification tasks, where they have demonstrated competitive performance Ma et al. (2024); Wang et al. (2021) compared to E2E methods by tailor-made auxiliary network. Despite this, the focus on auxiliary networks architecture for classification has constrained their general applicability. When extending these architectures to more complex tasks like object detection or super-resolution, they often fall short due to their lack of cross-task adaptability and the widely recognized "short-sightedness" problem Su et al. (2024b). Although the work Su et al. (2024a) mitigates short-sightedness by using exponential moving averages to enhance single-scale communication, it fails to address the deeper limitations posed by cross-task adaptability challenges, especially where multi-scale information is essential. For example, object detection requires multi-scale information, and the classification-oriented architecture's lack of it exacerbates the short-sightedness issue. Consequently, these issues limit the potential of traditional local learning methods, hindering their generalization and portability across diverse visual tasks.

To this end, we present the Memory-Augmented Network (MAN), a novel framework designed to address the challenges of scaling local learning methods across diverse tasks. This streamlined approach alleviates the above short-sightedness issue between local modules at different scales and enables performance that closely matches end-to-end training. Specifically, MAN operates as an

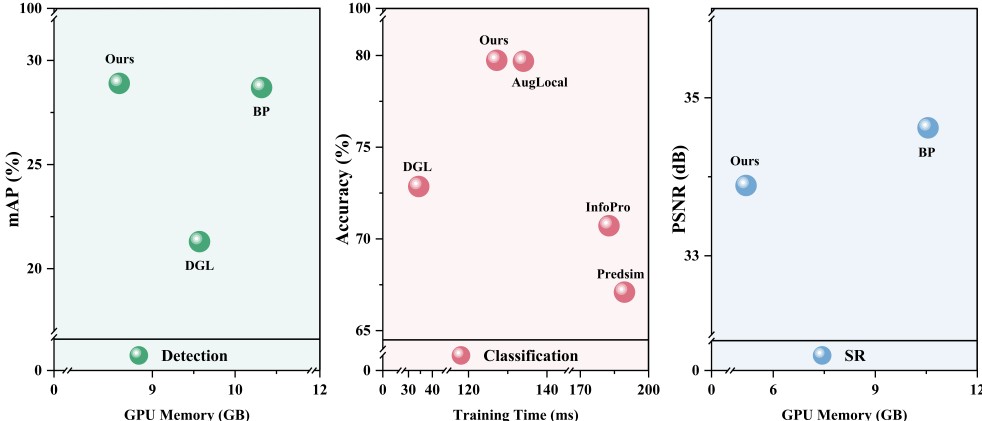

Figure 1: We compare with the state-of-the-art local learning and E2E methods on object detection, classification, and super-resolution tasks.

auxiliary network within each gradient-isolated local module, adapting automatically to the target task's architecture, thus eliminating the need for manual design adjustments. It features a straightforward local module that emphasizes the reusability of task-specific knowledge, facilitating the extension of local learning to various applications. A key innovation is the incorporation of a feature bank to enable multi-scale feature communication, allowing MAN to capture both generalized and discriminative semantic features. By integrating cross-scale information from the feature bank, MAN constructs a comprehensive semantic representation, advancing supervised local learning beyond classification tasks. Extensive experiments show that MAN achieves comparable performance to end-to-end methods on a variety of challenging tasks shown in Figure1, including image classification, object detection, and super-resolution, while significantly saving GPU memory.

The main contributions of this paper are as follows:

- This paper introduces the Memory-Augmented Network (MAN), which simplifies the network structure for corresponding tasks. By facilitating access to cross-scale features, it effectively addresses the needs of diverse applications, enabling the seamless extension of local learning.

- Comprehensive experiments on image classification, object detection, and super-resolution tasks validate the effectiveness of the MAN-designed local learning network. The MAN approach achieves performance comparable to end-to-end back-propagation (BP) while significantly reducing GPU memory usage.

- An in-depth analysis of the latent representations learned by models utilizing the MAN method reveals that, compared to BP, local networks enhanced with key global information help the network learn more discriminative features at shallow layers, thereby improving the overall performance of the model.

## 2 RELATED WORK

### 2.1 LOCAL LEARNING

Local learning is an innovative algorithm that strictly adheres to biological plausibility principlesCrick (1989), aimed at utilizing memory more effectively for deep learning. Its key advantage lies in addressing the limitations of global end-to-end (E2E) trainingHinton et al. (2006), thereby fostering the development of alternative supervised local learning methods. In supervised local learning, the progression of training primarily depends on using reasonable supervised local loss functions or constructing efficient manual auxiliary networks. Previous research in differentiable search algorithms utilizes self-supervised contrastive loss functions under local learning rulesIlling et al. (2021); Xiong et al. (2020); Nøkland & Eidnes (2019); Wang et al. (2021), enabling local

block-level learning through decoupling network blocks and selecting manual auxiliary networks for each blockPyeon et al. (2020); Wang et al. (2021); Belilovsky et al. (2020). However, when the aforementioned networks are partitioned into numerous local blocks without restraint, the performance of the network suffers significantly due to the inability of backpropagation to effectively relate parameters between these local blocks.

## 2.2 Alternative Methods to E2E Training

Due to the increasingly apparent shortcomings of E2E training, many researchersLillicrap et al. (2020) have diligently pursued alternative approaches to E2E training to address the areas in E2E training that did not adhere to biological plausibility principles. The weight transfer problemCrick (1989) is addressed by attempting to directly propagate the global error to each hidden unitNøkland (2016); Clark et al. (2021), or by employing different feedback connectionsLillicrap et al. (2016); Akrout et al. (2019). AndLee et al. (2015); Bengio (2014); Le Cun (1986) employ localized object reconstruction to train a dedicated backward-connected target propagation training method. while recent workRen et al. (2022); Dellaferrera & Kreiman (2022)creatively employed forward gradient learning to completely bypass the drawbacks of network backpropagation. These methods have, to some extent, strengthened the biological plausibility principles of the network. However, they struggle to achieve efficient performance on large datasetsDeng et al. (2009). Furthermore, their critical drawback—dependency on global objectives, remains unresolved, which is fundamentally different from the structure of neural systems in the real world that relies on local neuron connections to transmit and update information.

## 2.3 Object Detection

R-CNN and Faster R-CNNGirshick et al. (2014); Ren et al. (2015) are the origin and excellent succession of the classic R-CNN model, respectively. They employed simple and scalable networks for object detection, yet achieved very high detection accuracy. YOLOv1Redmon et al. (2016) and YOLOv8Jocher (2023) represent the pioneering work and the latest iteration of the YOLO (You Only Look Once) series, respectively. They treat object detection as a regression problem to spatially separated bounding boxes and associated class probabilities, making it a real-time, fast object detection model. On the other hand, RetinaNetLin et al. (2017b) is a dense detector utilizing focal loss, offering high detection accuracy. DETRCarion et al. (2020) simplified the detection process by directly treating object detection as a set prediction problem. This significantly reduced the need for many components. However, the aforementioned methods still face the issue of high memory consumption during training.

## 2.4 Image Super-Resolution

Image Super-Resolution (SR) research aims to reconstruct High-Resolution (HR) images from Low-Resolution (LR) images. This technology has significant applications in various fieldsWang et al. (2020); Georgescu et al. (2023); Razzak et al. (2023). SRCNNDong et al. (2015) is the pioneer of deep learning-based super-resolution models. It is a simple model that addresses the image restoration problem using just three layers, achieving impressive results. EDSRLim et al. (2017) is an enhanced deep super-resolution network that improves model performance by removing unnecessary modules from the traditional residual network. RCANZhang et al. (2018) and SwinIRLiang et al. (2021) utilize a very deep residual channel attention network and Swin Transformer, respectively, for high-precision image super-resolution. Both have achieved outstanding results in super-resolution tasks. However, the aforementioned image super-resolution models typically require a significant amount of computational resources and storage space. This is particularly problematic when handling high-resolution images, as they tend to consume excessive Graphics memory. This is an urgent challenge that needs to be addressed, and our research can effectively resolve this issue.

# 3 METHOD

## 3.1 PRELIMINARIES

To establish the foundation of our work, we begin with a brief overview of traditional end-to-end supervised learning and backpropagation mechanisms. Let $x$ denote a data sample and $y$ its corresponding ground-truth label. We partition the entire deep network into several local blocks. During forward propagation, the output of the $j$-th block becomes the input to the $(j + 1)$-th block, which can be expressed as $x_{j+1} = f_{\theta_j}(x_j)$. Here, $\theta_j$ represents the parameters of the $j$-th local block, and $f(\cdot)$ denotes the forward computation performed by the block. We compute the loss function $\mathcal{L}(\hat{y}, y)$ between the output $\hat{y}$ of the final block and the ground-truth label $y$, and then iteratively propagate this loss backward through the preceding blocks.

Supervised local learning strategies Nøkland & Eidnes (2019); Wang et al. (2021); Belilovsky et al. (2020) incorporate auxiliary networks to provide local supervision. In this approach, an auxiliary network is attached to each local block. The output of each local block is fed into its corresponding auxiliary network, generating a local supervisory signal expressed as $\hat{y}_j = g_{\gamma_j}(x_{j+1})$. Here, $\gamma_j$ denotes the parameters of the $j$-th auxiliary network.

In this setup, we update the parameters of the j-th auxiliary network and local block, $\gamma_j, \theta_j$, as follows:

$$\gamma_j \leftarrow \gamma_j - \eta_a \times \nabla_{\gamma_j} \mathcal{L}(\hat{y}_j, y) \tag{1}$$

$$\theta_j \leftarrow \theta_j - \eta_l \times \nabla_{\theta_j} \mathcal{L}(\hat{y}_j, y) \tag{2}$$

Here, $\eta_a$ and $\eta_l$ denote the learning rates of the auxiliary networks and the local blocks, respectively. By attaching auxiliary networks, each local block becomes gradient-isolated and can be updated using local supervision instead of global backpropagation.

However, when attempting to transfer existing local learning architectures to other tasks, significant performance gaps arise for two main reasons. First, traditional local learning methods rely on meticulously designed auxiliary network structures. While these structures enhance performance in classification tasks, they constrain the network design to specific tasks, hindering adaptability during transfer to other tasks. Second, Su et al. (2024b;a) highlight that a key limitation of local learning architectures is the short-sightedness problem in addressing local modules. Unlike classification tasks, other tasks often require features at different scales, which exacerbates the effects of short-sightedness and complicates task transfer.

## 3.2 MEMORY-AUGMENTED NETWORK

We propose the Memory-Augmented Network (MAN) architecture to extend local learning to different tasks. In this Chapter, we will briefly explain the MAN framework, and introduce how MAN can transfer local-learning to different tasks in detail in Chapter 4.

As shown in the Figure 2, MAN consists of Simple Local Modules (SLM) and a Feature Bank, each addressing two key challenges in task transfer. SLM provides a simple auxiliary network adaptable to various tasks, overcoming the challenge of task-specific designs. The feature bank stores multi-scale features and applies them in the auxiliary network, which alleviates the myopic problem of local modules. Together, these components simplify the transfer of local learning techniques across tasks. We will introduce these two parts separately

**SLM:** In networks designed for different tasks, the backbone is often carefully crafted based on the specific task, leading to significant differences in the feature extraction processes of different network architectures. By simply downsizing the backbone network, we retain its feature extraction capabilities and ensure that the auxiliary network aligns with the objectives of the backbone. This allows SLM to be easily adapted to different task requirements.

Through the design of SLM, local-learning can be easily extended to various tasks. However, this can not solve the short-sighted problem of local modules inherent in local-learning, resulting in the loss of task performance. We propose Feature bank to make up for this deficiency.

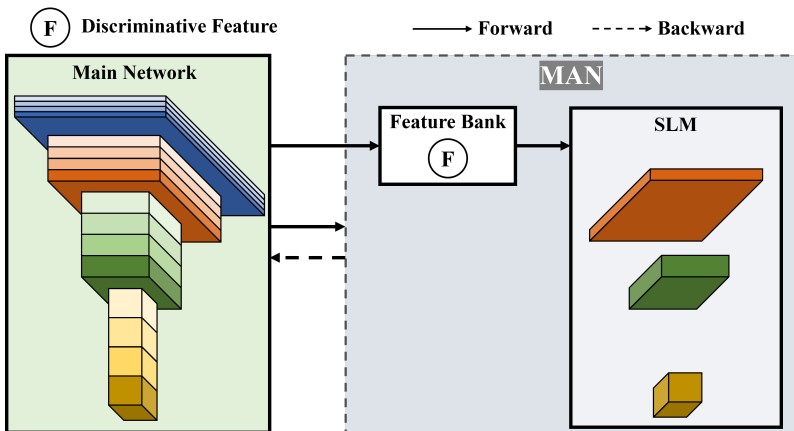

Figure 2: Structure diagram of MAN method, where MAN consists of Feature Bank and SLM. By extracting Discriminative features from the main network, and constructing the SLM network homogeneous with the main network, the local modules of MAN and the backbone network are used to update the gradient

**Feature Bank:** Feature bank is a repository for key features. In different tasks, there are some key features of their respective tasks, such as fc features in classification tasks, multi-scale features in target detectionLin et al. (2017a), and initial image features in low-level tasks represented by super-resolution tasksLim et al. (2017), which have been proved to play a decisive role in their respective tasks. The traditional lcoal-learning method has little influence on the model because the key feature of the classification task is the last classification layer. However, when moving to areas such as object detection, the absence of multi-scale features can have disastrous consequences, as shown in Table.3. We do this by storing these key features in the Feature bank and using them in local-modules just as they are used in the backbone network. To alleviate the problem of short-sightedness between different blocks, after making a certain memory sacrifice, in exchange for amazing performance.

The feature map produced by layer $l$ of the backbone network $\mathcal{B}$ as $\mathbf{F}^{(l)} \in \mathbb{R}^{h \times w \times d}$, where $h$, $w$, and $d$ represent the spatial height, width, and depth of the feature map, respectively.

We denote the set of distinct features extracted from various layers of $\mathcal{B}$ as the *feature bank*, $\mathcal{F}_{\text{bank}}$. The feature bank is formally defined as a set of distinct feature representations:

$$\mathcal{F}_{\text{bank}} = \left\{ \mathbf{F}_i^{(l)} \mid i \in I, l \in \{1, 2, \ldots, L\} \right\}, \tag{3}$$

where $I$ is an empirically derived index set of distinct feature maps and $\mathbf{F}_i^{(l)}$ represents a selected feature map from layer $l$.

## 4 APPLICATIONS

In different tasks, models are typically divided into a backbone network for feature extraction and a head network for task execution. This structure allows for easy transfer of an end-to-end (E2E) framework to other tasks by simply replacing the task-specific head. However, in local learning, the incomplete network structure complicates the straightforward transfer to other tasks. For instance, in object detection, predictions require features from different scales; however, gradient-independent local networks often miss other essential features, making accurate predictions challenging.

Next, we will elaborate on how the MAN framework is applied to different tasks. We have selected three classic tasks: classification, object detection, and super-resolution. They respectively represent traditional application scenarios of local learning, challenging tasks in high-level tasks, and the most representative tasks in low-level tasks.Due to the characteristics of classification networks, we will focus on the construction of the backbone network in classification tasks, as well as how to apply MAN to these tasks. In subsequent sections on detection and super-resolution, we will reuse the backbone network architecture and emphasize how to leverage MAN for task transfer.

### 4.1 MAN IN CLASSIFICATION

Taking the most classical ResNet classification network as an example, Its structure is shown in Figure 3.If $\mathcal{B}$ is a ResNet architecture of $L_\mathcal{B}$ stages.Each auxiliary network $A_i$ has a reduced structure corresponding to the architecture of the backbone network $\mathcal{B}$. where each stage $s$ has $n_s$ layers, then the auxiliary network $\mathcal{A}_i$ also contains $L_{\mathcal{A}_i}$ stages with reduced layers, such that:

$$L_{\mathcal{A}_i} = L_\mathcal{B}, \quad \text{and} \quad n_{s,\mathcal{A}_i} \ll n_s \quad \text{for all stages } s. \tag{4}$$

Specifically, we define $n_{s,\mathcal{A}_i} = 1$ for simplicity, which means that each stage in the auxiliary network $\mathcal{A}_i$ consists of a single layer

Each auxiliary network $\mathcal{A}_i$ computes its own independent loss using a loss function structure identical to that of the backbone network $\mathcal{B}$. Let $\mathcal{L}_\mathcal{B}$ be the loss function of the backbone network. The loss function for auxiliary network $\mathcal{A}_i$ is denoted as $\mathcal{L}_{\mathcal{A}_i}$ and has the same structure as $\mathcal{L}_\mathcal{B}$, but is computed on the output of $\mathcal{A}_i$ using features from the feature bank $\mathcal{F}_{\text{bank}}$. Formally, the loss for auxiliary network $\mathcal{A}_i$ is defined as:

$$\mathcal{L}_{\mathcal{A}_i} = \mathcal{L}_{\text{structure}} \left( \mathbf{y}_i, f_{\mathcal{A}_i}(\mathbf{x}; \mathcal{F}_{\text{bank}}, \mathbf{W}_i) \right), \tag{5}$$

where $\mathbf{y}_i$ represents the target for auxiliary network $\mathcal{A}_i$, $f_{\mathcal{A}_i}$ is the forward function of $\mathcal{A}_i$, and $\mathbf{W}_i$ is the set of weights for $\mathcal{A}_i$. The function $f_{\mathcal{A}_i}$ uses the distinct features from the feature bank $\mathcal{F}_{\text{bank}}$ as input along with the weights $\mathbf{W}_i$ to generate predictions.

Each auxiliary network $\mathcal{A}_i$ optimizes its loss $\mathcal{L}_{\mathcal{A}_i}$ independently from the backbone network and from other auxiliary networks:

$$\min_{\mathbf{W}_i} \mathcal{L}_{\mathcal{A}_i}, \quad \forall i \in \{1, 2, \ldots, N\}. \tag{6}$$

This ensures that each auxiliary network adapts its parameters solely based on its own unique feature set, preventing interference with other networks.

### 4.2 MAN IN OBJECT DETCETION

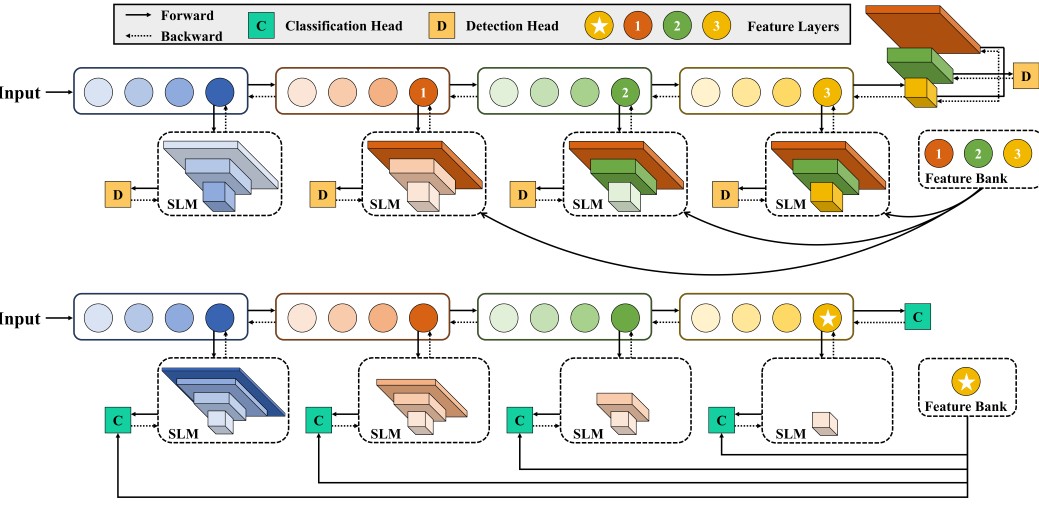

Figure 3: MAN is applied to structural diagrams in different ways, The dashed lines between layers represent the gradient feedback flow between layers, the network below the main network is the local network structure of the corresponding layer, and different colors represent the corresponding key features. Where, the top of the picture is MAN in Classification, and the bottom is Detection.

Figure 3 illustrates the overall architecture of MAN in object detection networks. The backbone network is divided into $K$ local modules, each equipped with a MAN. The parameters of each local

module are updated through its respective auxiliary network. Arrows between each layer and its auxiliary network represent the gradient flow, while arrows between different layers indicate the selection and utilization of features.

In object detection, we reuse the same method used in classification to simplify the feature extraction part of the backbone network. During the head design process, since studies have shown that multi-scale features are decisive for detection performance, we directly share the head part of the main network to train and understand multi-scale information more effectively. At the same time, the multi-scale FPN features are selected to be stored into the feature library. This allows the auxiliary network to build a local FPN network for outcome prediction.

At the beginning of training, the feature maps of different scales generated by the backbone network are stored into the feature library. We select feature maps from the layers preceding each down-sampling layer to ensure information richness. In different local modules, we use SLM to extract the multi-scale features needed to synthesize local FPN in the local network. At the same time, the multi-scale features required by the global FPN are saved in the Feature bank, and the local FPN is synthesized by reusing the feature maps existing in the feature bank. The design is adaptable and avoids the need for manual design of auxiliary networks.

### 4.3 MAN IN SUPER-RESOLUTION

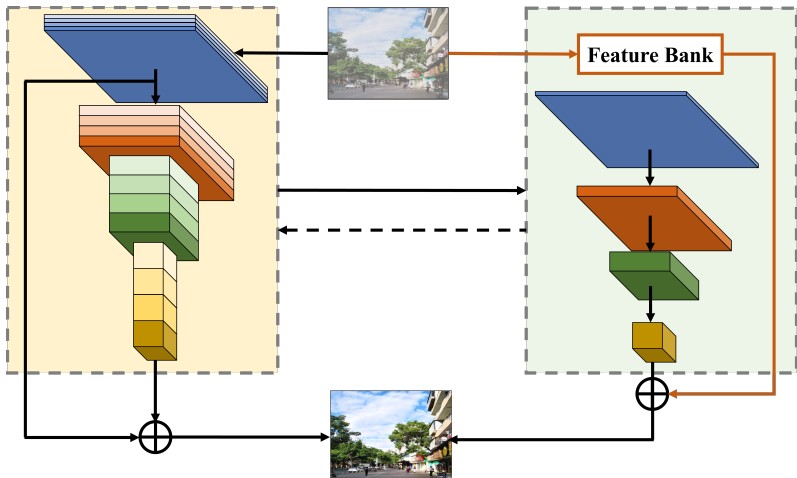

Figure 4: MAN in SR structure diagram, the left is the main network, the green structure is the corresponding local-module of this layer, and its specific structure is the part in the right box in the figure. The key feature is the initial input image, whose flow pattern is shown along the orange line.

Figure 4 illustrates the network architecture for applying MAN to the super-resolution domain. Similar to what we did in object detection, we build an SLM network that can be used for super-resolution by reusing the simplified way of MAN in the backbone network and adding the head required for super-resolution tasks. At the same time, we build MAN by finding discriminative features in super-resolution tasks and putting them into the Feature bank.

The core of super-resolution task is to preserve the original image features during the reconstruction process. A key approach is to fuse the feature map with the initial low-resolution image before upsampling, since the initial image is crucial for reconstructing the final high-resolution image. We store the initial image in the Feature bank and use it for reconstruction when each Local modules predicts imaging.

## 5 EXPERIMENTS

In this section, we conduct experiments to evaluate the performance of the MAN architecture on different tasks. Section 5.1 presents the performance of MAN on traditional classification tasks,

where we conduct experiments on CIFAR-10 Krizhevsky et al. (2009), SVHN Netzer et al. (2011), and STL-10 Coates et al. (2011). We select four state-of-the-art supervised local learning methods for comparison: PredSim Nøkland & Eidnes (2019), DGL Belilovsky et al. (2020), InfoPro Wang et al. (2021), and AugLocal Ma et al. (2024).In Section 5.2, we conduct more comprehensive experiments on object detection tasks, providing an in-depth comparison to highlight the performance of MAN in the field of object detection. We perform quantitative evaluations on the VOC Everingham et al. (2010) and COCO Lin et al. (2014) datasets, achieving results close to end-to-end (E2E) methods. We also compare local detection with other advanced local learning methods in terms of reducing memory overhead. An ablation study on the architecture is conducted, providing in-depth insights and qualitative analyses.In Section 5.3, we apply MAN to the super-resolution task and validate its performance on the widely used DIV2K Agustsson & Timofte (2017) dataset. The detailed experimental settings can be found in Appendix A.

Table 1: Results on the Classification

| Method | CIFAR-10 | | STL-10 | | SVHN | |
|--------|------|-----------------|-------|-----------------|-------|-----------------|
| | ACC | Inference Speed | ACC | Inference Speed | ACC | Inference Speed |
| Predsim | 77.29 | 174.50 | 67.1 | 189.40 | 91.92 | 174.30 |
| DGL | 85.92 | 48.30 | 72.86 | 34.30 | 94.95 | 43.60 |
| InfoPro | 87.07 | 192.10 | 70.72 | 182.60 | 94.03 | 197.70 |
| AugLocal | 92.48 | 90.60 | 79.69 | 134.00 | 96.80 | 111.80 |
| Ours | 91.75 | 81.70 | 79.74 | 127.20 | 96.68 | 90.00 |

## 5.1 Classification Experimental Results

We first evaluate the performance of the MAN method on classic image classification tasks. As shown in Table 1, although achieving the best performance was not our primary goal, MAN still obtains the best or second-best results on the CIFAR-10, STL-10, and SVHN datasets, indicating that it maintains strong transferability without sacrificing performance. Notably, in terms of inference speed, due to the simplicity of the SLM design, we avoid using complex auxiliary network structures, allowing MAN to achieve remarkable speed, second only to DGL. However, DGL's accuracy is significantly lower than MAN's, highlighting MAN's superior balance between performance and efficiency.

## 5.2 Object Detection Experimental Results

**Results on VOC dataset:** To verify the performance of the MAN method, we first conduct experiments on VOC dataset using the traditional Back Propagation (BP) method with our MAN. The experimental results are shown in Table 2. Surprisingly, the MAN method achieves comparable performance to the BP method in the vast majority of experimental groups. It is worth noting that the MAN method achieves higher mAP results in the experiments with RetinaNet-R50 and other models. This improvement is attributed to our model scoring better on smaller objects such as bottles and cars. This may be due to the fact that the MAN method can help the model identify the focal features earlier, while the shallow layers used to identify small objects can effectively learn more discriminative features. This leads to an overall performance enhancement of the model.

Moreover, we observe that even though the mAP performance of the MAN method is comparable to that of the BP method, its $AP_{50}$ and $AP_{75}$ scores are still lower than those of the BP method. This suggests that at higher threshold Settings, the features learned by the MAN method may be more discriminative than the BP method, leading to better performance at these thresholds.

**Result on MS COCO:** We evaluate the performance of our MAN method on the more challenging MS COCO dataset Lin et al. (2014). To control experimental costs and considering the fast training speed of DGL shown in Table 1, we conduct experiments using RetinaNet-R34 for the DGL, MAN, and BP methods. For traditional local-learning methods like DGL, we build an FPN structure in the auxiliary network for DGL through upsampling and other transformations to realize transfer, instead of simply predicting directly. This enhancement aims to provide better performance for traditional

Table 2: Results on the validation set of VOC.

| Model | mAP | aero | bike | bird | boat | bottle | bus | car | cat | chair | cow | table | dog | horse | mbike | person | plant | sheep | sofa | train | tv |
|---|---|---|---|---|---|---|---|---|---|---|---|---|---|---|---|---|---|---|---|---|---|
| RetinaNet-R34 | 53.9 | 68.6 | 58.2 | 49.4 | 40.3 | 21.5 | 58.2 | 50.7 | 81.6 | 31.9 | 51.0 | 43.8 | 76.9 | 71.5 | 65.3 | 54.8 | 22.4 | 43.2 | 55.6 | 76.6 | 57.3 |
| Ours (K=17) | 52.2 | 64.8 | 55.6 | 44.5 | 39.7 | 24.1 | 58.6 | 56.3 | 83.4 | 28.6 | 49.8 | 37.4 | 66.3 | 69.4 | 58.9 | 57.3 | 22.6 | 36.8 | 53.9 | 80.6 | 51.4 |
| RetinaNet-R50 | 56.2 | 67.5 | 59.6 | 53.1 | 44.8 | 24.1 | 58.2 | 54.5 | 82.6 | 30.7 | 57.8 | 44.5 | 80.9 | 76.8 | 68.3 | 56.5 | 21.2 | 46.5 | 57.9 | 79.1 | 60.2 |
| Ours (K=17) | 56.5 | 64.9 | 51.3 | 56.5 | 45.2 | 25.3 | 58.9 | 55.7 | 85.1 | 29.3 | 58.4 | 40.5 | 73.9 | 76.6 | 64.9 | 59.2 | 22.9 | 37.5 | 56.5 | 83.2 | 59.0 |
| RetinaNet-R101 | 58.2 | 69.9 | 61.6 | 53.0 | 51.5 | 26.1 | 61.7 | 57.1 | 84.3 | 35.5 | 58.7 | 44.4 | 81.4 | 77.1 | 70.4 | 58.7 | 24.0 | 45.8 | 61.7 | 81.8 | 60.8 |
| Ours (K=34) | 56.9 | 62.4 | 55.5 | 57.1 | 50.9 | 25.5 | 61.4 | 55.9 | 86.7 | 32.4 | 57.3 | 39.9 | 77.1 | 77.4 | 65.5 | 59.4 | 25.2 | 35.5 | 57.9 | 83.9 | 56.5 |
| RetinaNet-R152 | 61.0 | 72.2 | 64.8 | 57.7 | 50.2 | 31.9 | 62.8 | 59.9 | 85.3 | 41.2 | 63.2 | 53.3 | 81.9 | 78.9 | 70.2 | 61.2 | 28.5 | 47.4 | 63.2 | 81.5 | 65.0 |
| Ours (K=51) | 60.9 | 69.4 | 60.5 | 62.7 | 51.1 | 30.5 | 65.1 | 56.4 | 83.5 | 43.3 | 60.7 | 54.0 | 74.9 | 81.4 | 62.8 | 63.7 | 27.1 | 45.5 | 63.1 | 85.5 | 65.6 |
| YOLO-R34 | 58.9 | 63.6 | 65.2 | 62.9 | 42.2 | 30.6 | 67.7 | 67.4 | 77.3 | 36.4 | 63.5 | 49.9 | 74.3 | 76.8 | 67.5 | 60.6 | 27.4 | 60.0 | 52.2 | 72.9 | 60.3 |
| Ours (K=17) | 58.6 | 64.2 | 63.5 | 63.3 | 34.8 | 30.1 | 66.9 | 65.1 | 77.3 | 30.5 | 62.8 | 48.5 | 70.3 | 82.6 | 65.4 | 61.0 | 28.1 | 61.2 | 49.7 | 72.4 | 61.3 |
| YOLO-R50 | 58.5 | 57.0 | 73.2 | 60.9 | 37.8 | 30.4 | 66.6 | 66.5 | 76.7 | 37.7 | 61.1 | 44.2 | 76.9 | 77.0 | 67.8 | 60.1 | 29.3 | 58.8 | 56.5 | 64.6 | 60.8 |
| Ours (K=17) | 57.1 | 56.9 | 73.7 | 59.4 | 35.5 | 31.8 | 61.4 | 67.8 | 77.9 | 34.5 | 60.4 | 44.4 | 68.5 | 81.8 | 66.1 | 61.9 | 29.1 | 54.3 | 55.9 | 70.1 | 60.9 |
| YOLO-R101 | 60.4 | 65.1 | 68.1 | 64.9 | 45.1 | 27.5 | 69.1 | 68.2 | 76.7 | 38.6 | 65.0 | 51.1 | 76.6 | 78.8 | 73.4 | 62.5 | 32.4 | 62.2 | 57.2 | 67.7 | 57.1 |
| Ours (K=34) | 60.6 | 65.7 | 64.5 | 62.9 | 47.4 | 29.1 | 67.1 | 70.7 | 78.4 | 36.9 | 64.3 | 53.4 | 70.2 | 79.9 | 74.2 | 61.8 | 33.5 | 59.9 | 57.5 | 70.5 | 55.8 |
| YOLO-R152 | 64.3 | 66.2 | 77.6 | 68.9 | 50.3 | 38.1 | 69.9 | 75.1 | 76.8 | 43.8 | 72.5 | 51.0 | 77.8 | 81.2 | 74.8 | 63.8 | 34.6 | 64.8 | 57.0 | 74.8 | 66.2 |
| Ours (K=51) | 63.5 | 65.9 | 70.7 | 64.5 | 50.1 | 42.3 | 65.7 | 73.9 | 80.1 | 44.1 | 65.8 | 52.2 | 74.2 | 80.8 | 76.0 | 64.4 | 36.5 | 59.4 | 55.5 | 77.9 | 56.1 |

Table 3: Results on the validation set of COCO.The red arrow represents the accuracy improvement of the MAN method compared with the traditional Local-learning on the detectin task, and the green arrow represents the ability of the MAN method to save memory overhead compared with the BP method

| Model | Method | $mAP$ | $AP_{50}$ | $AP_{75}$ | GPU Memory |
|---|---|---|---|---|---|
| RetinaNet-R34 | BP | 28.7 | 49.3 | 29.5 | 10.32GB |
|  | DGL(K=8) | 21.3 | 43.9 | 16.6 | 9.57GB |
|  | Ours(K=8) | 28.9(↑7.6) | 48.7(↑4.8) | 28.7(↑12.1) | 8.60GB (↓16.70%) |
|  | DGL(K=17) | 19.5 | 41.8 | 15.2 | 9.37GB |
|  | Ours(K=17) | 28.5(↑9.0) | 47.6(↑5.8) | 28.4(↑13.2) | 8.19GB (↓20.60%) |
| RetinaNet-R50 | BP | 29.7 | 51.6 | 30.4 | 18.05GB |
|  | Ours(K=17) | 29.4 | 48.3 | 29.7 | 15.34GB (↓15.01%) |
| RetinaNet-R101 | BP | 31.8 | 53.6 | 32.3 | 22.46GB |
|  | Ours(K=34) | 31.9 | 52.4 | 32.6 | 20.61GB (↓8.24%) |
| RetinaNet-R152 | BP | 33.2 | 56.2 | 33.5 | 31.84GB |
|  | Ours(K=51) | 33.5 | 56.1 | 33.6 | 28.36GB (↓10.92%) |
| YOLO-R34 | BP | 20.23 | 41.27 | 21.10 | 11.80GB |
|  | Ours(K=17) | 20.16 | 40.38 | 20.06 | 8.89GB (↓24.66%) |
| YOLO-R50 | BP | 20.94 | 42.02 | 21.97 | 26.15GB |
|  | Ours(K=17) | 20.97 | 42.00 | 20.15 | 21.37GB (↓18.27%) |
| YOLO-R101 | BP | 22.41 | 44.36 | 22.53 | 37.05GB |
|  | Ours(K=34) | 22.35 | 43.67 | 22.18 | 26.40GB (↓28.77%) |
| YOLO-R152 | BP | 25.00 | 47.15 | 24.33 | 49.88GB |
|  | Ours(K=51) | 24.84 | 45.34 | 24.91 | 40.02GB (↓19.76%) |

local-learning methods, ensuring a fair comparison. However, as shown in Table 3, there remains a significant performance gap between the DGL method and the BP method.Our experiments indicate that although DGL structurally possesses the capability to perform object detection tasks, the absence of multi-scale information hinders its performance in extending to such tasks effectively.

Notably, in the RetinaNet-R34 experiments, when $K < 8$, and in the RetinaNet-R101 experiments, MAN outperforms BP, confirming MAN's significant potential. It can also be observed that as the number of segments $K$ decreases, the model performance generally improves. However, when $K = 4$, MAN's mAP performance slightly decreases, with the $AP_{50}$ metric increasing and the $AP_{75}$ metric decreasing. This further supports our hypothesis that, compared to BP, local learning methods can help the model learn different feature representations.

When comparing GPU memory usage, MAN demonstrates superior memory-saving capabilities compared to DGL due to our simplified local structure. In RetinaNet-R34, YOLO-R34, and YOLO-R101, MAN reduces memory overhead by 20.6%, 24.66%, and 28.77%, respectively, while maintaining performance comparable to BP.

**Ablation Study:** We perform ablation experiments on MAN; due to space limitations, we only present part of the experiments in the main text, and other experiments will be provided in the supplementary material.

We try to incrementally reduce the modules of the local detection, and the results are shown in Table 4. Where Adapt represents whether to use MAN's SLM and Feature bank methods, and Head represents whether to share the same detection head with the network. We find that although the shared detection head can help the model improve the performance at a certain increase in memory overhead. This shows that while one can simply introduce local learning methods to the task, there is still much room for improvement in how to exploit these important features once they are added to the local network. There may be potential to consistently outperform BP architectures. But achieving state-of-the-art performance in each task is not the main goal of this paper; We leave it as future work.

Table 4: Ablation study between modules of different local detection schemes. Here, Adapt indicates whether the adaptive MAN network is used, and Head indicates whether the shared prediction head is used.

| Adapt | Head | mAP | GPU Memory |
|-------|------|------|-----------|
| × | × | 28.7 | 10.32 |
| ✓ | × | 27.7 | 8.07 |
| ✓ | ✓ | 28.5 | 8.19 |

Table 5: Results on the validation set of DIV2K.

| Task | Method | PSNR | GPU Memory |
|------|--------|-------|-----------|
| ×2 | BP | 34.62 | 10.55GB |
|    | Ours | 33.89 | 5.20GB |
| ×3 | BP | 31.04 | 10.30GB |
|    | Ours | 29.33 | 5.19GB |
| ×4 | BP | 28.92 | 10.06GB |
|    | Ours | 27.71 | 5.16GB |

## 5.3 SUPER-RESOLUTION EXPERIMENTAL RESULTS

We conduct experiments on the DIV2KAgustsson & Timofte (2017) dataset to evaluate the performance of our model. Because the traditional Local-learning method lacks the key information of the initial image, it leads to a catastrophic performance gap and is difficult to be transferred to super-resolution tasks. We choose to perform a more nuanced comparison with the E2E method. As shown in Table 5, our MAN method shows a performance gap compared to the BP method, with a difference of 1.73 in the ×2 task,1.71 in the ×3 task and 1.21 in the ×4 task.It can be observed that the performance gap between MAN and BP is gradually narrowing as the difficulty of the super-resolution task increases, which may be due to the fact that MAN is able to learn more essential features. At the same time, the GPU overhead of the MAN method is only 5.20GB, 5.19GB and 5.16GB, saving 51% of memory. This may be due to the fact that the super-resolution task, apart from the backbone, is simpler compared to the object detection task, which does not involve complex head components, leading to a smaller proportion of additional memory overhead in the model.

## 6 CONCLUSION

In this paper, we introduce Memory-Augmented Network (MAN), a novel design of auxiliary networks that for the first time extends the application of local learning to different tasks. The design of MAN eliminates the need for tedious manual configuration and instead makes full use of the structure of the backbone network as well as the features at different levels. By augment the feature memory and increasing the utilization of cross-scale information by local modules, we apply MAN to different tasks, such as object detection and image super-resolution, and show that MAN significantly reduces GPU memory usage while maintaining comparable performance to BP.

**Limitations and Future Work:** Although the proposed Memory Augmented Network (MAN) performs well in terms of performance and adaptability for various tasks, it uses an explicit Feature bank, which brings additional memory overhead. Our future work will explore how to transfer information across scales implicitly.

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

# A    APPENDIX

## A.1    MORE RESULTS

**Representation Similarity:** We conduct a Centered Kernel Alignment (CKA)Kornblith et al. (2019) experiment to validate the effectiveness of MAN. Specifically, we calculate the CKA similarity for each layer using MAN, DGLBelilovsky et al. (2019), and BP under different methods and averaged results. As shown in Figure 5, the representation differences among the methods are minimal in the later layers, with DGL being closer to BP than MAN. However, MAN achieves higher similarity in the early layers due to its Focal Features Selection, which aids the model in learning key discriminative features early on. This experiment confirmed that MAN's design enhances the model's understanding of early features.

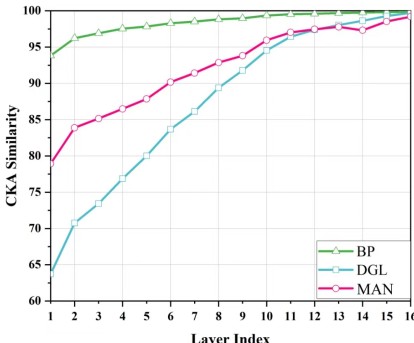

Figure 5: Diagram of the construction method of Memory-Augmented Network

## A.2    IMPLEMENTATION DETAILS OF CLASSIFICATION

In our experiments, we continue the same experimental setup as Auglocal. The experiments on CIFAR-10 Krizhevsky et al. (2009), SVHN Netzer et al. (2011), and STL-10 Coates et al. (2011) datasets with ResNet-32He et al. (2016), we utilize the SGD optimizer with Nesterov momentum set at 0.9 and an L2 weight decay factor of 1e-4. We employ batch sizes of 1024 for CIFAR-10 and SVHN and 128 for STL-10. The training duration spans 400 epochs, starting with initial learning rates of 0.8 for CIFAR-10 / SVHN and 0.1 for STL-10, following a cosine annealing scheduler Coates et al. (2011).

## A.3    IMPLEMENTATION DETAILS OF OBJECT DETECTION

**Dataset:** To validate the model's ability to fit large datasets, we use the VOC detection datasetEveringham et al. (2010) containing 9,963 images and the COCO datasetLin et al. (2014) containing 123,287 images for our object detection experiments. Additionally, all backbones are pre-trained on the ImageNet dataset, which includes approximately 1.3 million images.

**Model Variants:** To validate the scalability of the proposed method, we employ entirely different network architectures, namely YOLO Redmon et al. (2016) and RetinaNet Lin et al. (2017c). For a fair comparison with other models, the YOLO model used ResNet-based YOLOv1. Networks using the local detection method are referred to as MAN versions. Each model was trained using ResNet-34, ResNet-50, ResNet-101, and ResNet-152.

Furthermore, to compare the performance of the local detection method with other local learning methods in terms of memory overhead reduction, we conduct comparisons under the same model partitioning conditions. We adopted the state-of-the-art local learning method DGL Belilovsky et al. (2019) for the object detection task. To validate the effectiveness of the local detection algorithm, we compared its memory-saving performance.

**Training and Fine-tuning:** We utilize the SGD optimizer Keskar & Socher (2017) with Nesterov momentum Dozat (2016) set at 0.9 and an L2 weight decay factor of 1e-4. The training duration spans 160 epochs, with a learning rate employing a warm-up strategy that is set to 0 for the first 5 iterations, followed by 1e-4, and adheres to a cosine annealing schedule. When using ResNet-34 as the backbone, it is divided into 16 modules. Similarly, when employing ResNet-50, ResNet-101, and ResNet-152 as backbones, the networks are divided into 16, 33, and 50 modules, respectively. This division is based on the block parameters used in the construction of ResNet, with each local module's auxiliary network having its unique parameters. During training, RetinaNet uses a batch size of 64, whereas YOLO uses a batch size of 32.

### A.4 IMPLEMENTATION DETAILS OF SUPER-RESOLUTION

**Dataset:** For the super-resolution task, we utilize the DIV2K Agustsson & Timofte (2017) dataset, which comprises over 1000 high-resolution images, each exceeding 2K in resolution. This dataset is extensively employed in various super-resolution challenges and competitions.

**Model Variants:** On the DIV2K dataset, we conduct tests for 2x, 3x, and 4x super-resolution tasks to evaluate the model's performance, using EDSR as the benchmark model. The configurations employing the MAN method for local learning are denoted as EDSR-MAN.

**Training and Fine-tuning:** During training, we use patches of 48x48 low-resolution (LR) images and their corresponding high-resolution (HR) patches. ADAM is used as the optimizer, with the learning rate set at 1e-4. Initially, we begin training from scratch on the ×2 model. Once the model converges, it is used as a pre-trained network for training on the ×3 and ×4 models.

