# OpenReview forum: "Advancing Supervised Local Learning Beyond Classification with Long-term Feature Bank"
_ICLR.cc/2025/Conference — ICLR 2025 Conference Withdrawn Submission_

### Official Review · Reviewer_ukFa · 2024-11-02

**Soundness:** 3
**Presentation:** 2
**Contribution:** 2
**Rating:** 3
**Confidence:** 4

**Summary:**

The paper proposes the Memory-augmented Auxiliary Network (MAN), a framework that extends supervised local learning methods beyond image classification to other visual tasks such as object detection and super-resolution. This is achieved by addressing two main challenges: the lack of reusability of task-specific knowledge and the absence of cross-scale feature communication. The authors demonstrate the successful application of local learning methods beyond classification, showing that MAN conserves GPU memory and achieves performance comparable to end-to-end approaches across multiple datasets and visual tasks.

**Strengths:**

1. MAN shows promising results in extending the application of local learning to various visual tasks beyond classification.
2. It reduces GPU memory usage, which is beneficial for large-scale deep-learning applications where memory is a constraint.

**Weaknesses:**

1. The novelty is limited from my perspective. The design of SLM and Feature Bank is not novel enough for publishing.
2. I think the authors should make some efforts to discuss the local learning's advantages over BP DNN to demonstrate the motivation. Currently, from the authors' introduction, I do not see the necessity of replacing the BP with local learning in various visual tasks. I would suggest the authors to discuss thoroughly about this point, and make a detailed explanation of the model designs.
3. The additional feature bank does not contribute a lot to the results, while introducing extra computational overheads.
4. The model has not demonstrated the effectiveness in large-scale datasets like ImageNet.

**Questions:**

1. A lot of minors, like Line 103:  trainingHinton et al. Line 118: researchersLillicrap et al. Line 120: weight transfer problemCrick. etc.., where the spaces are missing. Please consider modifying these.
2. I would suggest that in Related Works, the OBJECT DETECTION and  IMAGE SUPER-RESOLUTION are not necessary, which the authors could consider moving them to appendix.

---

### Official Review · Reviewer_z9gs · 2024-11-04

**Soundness:** 3
**Presentation:** 3
**Contribution:** 2
**Rating:** 3
**Confidence:** 4

**Summary:**

The paper introduces the Memory-Augmented Network (MAN), an architecture designed to expand supervised local learning applications beyond traditional classification tasks. MAN uses a Memory-augmented Auxiliary Network, integrating a feature bank to enable cross-scale feature communication, which addresses short-sightedness in local modules and facilitates task adaptability. This framework allows supervised local learning to achieve comparable performance to end-to-end methods on multiple visual tasks, including classification, object detection, and super-resolution, with the added benefit of reduced GPU memory usage.

**Strengths:**

1. MAN demonstrates practical utility by achieving competitive performance across various visual tasks with reduced GPU memory usage, showing that it can offer a viable alternative to end-to-end backpropagation for resource-constrained scenarios.
2. MAN's straightforward auxiliary network design allows for adaptability across multiple tasks, effectively extending local learning to object detection and super-resolution. This approach removes the complexity of task-specific manual adjustments, which is advantageous for practical applications.
3. The paper includes detailed experimental validation, showing consistent performance improvements over existing local learning methods on benchmark datasets like CIFAR-10, VOC, COCO, and DIV2K.
4. By incorporating a feature bank, the paper addresses the limitations of traditional local learning methods, particularly the short-sightedness problem, enhancing the reusability of multi-scale feature representations.

**Weaknesses:**

1. While the memory component of MAN is intuitive and provides an engineering solution to enhance local learning, it lacks theoretical grounding or any biological mechanism correlation. This lack of deeper justification and analysis may limit the perceived significance of the approach, making it seem somewhat trivial as a method for overcoming local learning challenges.

2. The authors claim this work as the first successful application of local learning beyond classification, yet previous works, such as InfoPro, have already demonstrated substantial success in tasks like segmentation. This overstatement could be seen as a lack of rigor in contextualizing the novelty of the proposed approach.

3. The related work section and experimental comparisons lack several significant local training methodologies. For instance, recent works like Yang et al.'s "Towards Interpretable Deep Local Learning with Successive Gradient Reconciliation" (ICML 2024) present relevant advances in local learning, which could provide a meaningful benchmark against which MAN’s performance might be evaluated. This oversight diminishes the completeness of the analysis.

4. The paper restricts its evaluation to image-based tasks and confines the application of the feature bank to image-layer features. This narrow scope limits the potential generalizability of the proposed method to other domains, such as language or multimodal tasks, reducing the broad applicability of MAN in various local learning scenarios.

5. The authors only evaluate MAN's feature bank with backbones involving multi-scale features (e.g., ResNet, Swin-Transformer), overlooking models without pyramid structures, such as ViT for classification or DETR for object detection. This limited evaluation scope raises questions about the broader applicability of the feature bank approach, particularly for architectures that do not rely on multi-scale features.

**Questions:**

1. Could the authors elaborate on the theoretical basis for the feature bank, including its alignment (or lack thereof) with biological mechanisms in memory-augmented learning, and discuss why this approach is effective within the context of local learning?

2. Given that InfoPro has been successfully applied to segmentation tasks, how do the authors position MAN’s contributions relative to prior applications of local learning beyond classification, particularly in terms of novelty and advancement?

3. Could the authors provide a comparison with recent advancements in local learning, specifically Yang et al. (ICML 2024), to substantiate the effectiveness and relevance of MAN in light of alternative methods?

4. Has the potential generalizability of MAN to non-image tasks, such as text or multimodal scenarios, been considered? What modifications or evaluations would be required to test the adaptability of MAN beyond image-based tasks?

5. For models without inherent multi-scale structures, such as ViT or DETR, could the authors discuss potential adaptations of the feature bank or suggest experimental validations that might confirm MAN’s effectiveness in such architectures?

---

### Official Review · Reviewer_Nuo3 · 2024-11-04

**Soundness:** 3
**Presentation:** 2
**Contribution:** 2
**Rating:** 3
**Confidence:** 4

**Summary:**

The paper proposes the Memory-augmented Auxiliary Network, which employs a simplified design principle and integrates a feature bank to improve cross-task adaptability and communication, and provide experiment resuls in several downstream tasks to demonstrate the effectiveness of MAN.

**Strengths:**

The paper evalutated the proposed method on various tasks including classification, object detection and image super-resolution tasks among different datasets, show a good trade off between performance and inference speed, and represents the MAN's ability to save memory overhead.

**Weaknesses:**

The writing of the paper needs improvement. There are multiple spelling errors and unclear expressions throughout the article, such as "L242:We do this by storing these key features in the Feature bank and using them in local-modules just as they are used in the backbone network. To alleviate the problem of short-sightedness between different blocks, after making a certain memory sacrifice, in exchange for amazing performance."
2: The contribution/innovation of the paper is limited. The method is simple, the definition and description of the designed MAN network are unclear, and the results are not significant.
3: The practicality of the method is poor. Since the goal is to save storage space or improve inference efficiency, its effectiveness should be validated on larger network structures. Most GPUs now have over 12GB of memory, so reducing the memory usage of a small model from around 10GB to even less is unnecessary.
4. Is there an error in Figure 3? The figure shows dashed arrows between local modules. Is there a backpropagation signal between them? If so, how is it different from E2E backpropagation?

**Questions:**

The questions have been listed in the weakness.

---

### Note · Authors · 2024-11-14

I have read and agree with the venue's withdrawal policy on behalf of myself and my co-authors.